evolution, plant science

phenotypic plasticity, carnivorous plants, hidden reaction norm

**Authors for correspondence:**
Kenji Fukushima
e-mail: kenji.fukushima@uni-wuerzburg.de
Mitsuyasu Hasebe
e-mail: mhasebe@nibb.ac.jp

# A discordance of seasonally covarying cues uncovers misregulated phenotypes in the heterophyllous pitcher plant *Cephalotus follicularis*

Kenji Fukushima[1,2,3], Hideki Narukawa[1], Gergo Palfalvi[1,2] and Mitsuyasu Hasebe[1,2]

[1]National Institute for Basic Biology, Okazaki 444-8585, Japan
[2]Department of Basic Biology, School of Life Science, SOKENDAI (Graduate University for Advanced Studies), Okazaki 444-8585, Japan
[3]Institute for Molecular Plant Physiology and Biophysics, University of Würzburg, Julius-von-Sachs Platz 2, 97082 Würzburg, Germany

KF, 0000-0002-2353-9274; MH, 0000-0001-7425-8758

Organisms withstand normal ranges of environmental fluctuations by producing a set of phenotypes genetically programmed as a reaction norm; however, extreme conditions can expose a misregulation of phenotypes called a hidden reaction norm. Although an environment consists of multiple factors, how combinations of these factors influence a reaction norm is not well understood. To elucidate the combinatorial effects of environmental factors, we studied the leaf shape plasticity of the carnivorous pitcher plant *Cephalotus follicularis*. Clonally propagated plants were subjected to 12-week-long growth experiments in different conditions controlled by growth chambers. Here, we show that the dimorphic response of forming a photosynthetic flat leaf or an insect-trapping pitcher leaf is regulated by two covarying environmental cues: temperature and photoperiod. Even within the normal ranges of temperature and photoperiod, unusual combinations of the two induced the production of malformed leaves that were rarely observed under the environmentally typical combinations. We identified such cases in combinations of a summer temperature with a short-to-neutral day length, whose average frequency in the natural *Cephalotus* habitats corresponded to a once-in-a-lifetime event for this perennial species. Our results suggest that even if individual cues are within the range of natural fluctuations, a hidden reaction norm can be exposed under their discordant combinations. We anticipate that climate change may challenge organismal responses through not only extreme cues but also through uncommon combinations of benign cues.

## 1. Introduction

Phenotypic plasticity allows a single genotype to produce a set of phenotypes called a 'reaction norm' [1] in response to external cues. The induced phenotypes are considered a source of novel evolutionary traits accessed through genetic assimilation and accommodation, the evolutionary processes involving decreased or increased environmental sensitivity [2]. Latent but conspicuously different phenotypes can be exposed if an organism experiences unusually extreme growth conditions, which is called a hidden reaction norm [3]. Although the hidden reaction norm has been implicated as a potential source of evolutionary novelty [3], how it is exposed is not well understood.

Organisms must efficiently withstand the huge parameter space generated by exponential combinations of different environmental cues. In a plastic

developmental process, different environmental variables can be perceived as cues that trigger phenotypic plasticity. The perception of multiple cues has been generally thought to improve the accuracy of differential responses in biological systems [4,5], including phenotypic plasticity [6]. Previous works revealed that organisms can process multiple cues to regulate phenotypic plasticity by integrating them in various ways [7–9]. Under conditions typical of a habitat, highly auto-correlated environmental cues often change concurrently [10], as observed in the seasonal covariation of temperature and photoperiod. This fact highlighted a potential role for environmental covariation in regulating phenotypic plasticity and allowed us to develop a testable prediction that a discordance of covarying cues can expose hidden reaction norms.

To test this hypothesis, we focused on phenotypic plasticity in the carnivorous pitcher plant *Cephalotus follicularis*, which is endemic to the south-western corner of Western Australia [11]. Due to their low motility, phenotypic plasticity is likely to be an especially important adaptive trait in plants, which regularly experience environments that are heterogeneous in space and time [12]. Frequently, such plasticity is observed as variation in leaf forms and shapes where it is termed heterophylly [13]. Heterophylly is observed in various lineages of land plants, typically in aquatic plants [14] but also in terrestrial species [15], suggesting that it plays a broad role in plant adaptation. Various environmental cues, such as submergence, temperature, photoperiod, light intensity, light quality, and humidity, trigger heterophyllous responses [14,16]. In contrast to textbook examples of animal plasticity, such as predator-induced morphs in water fleas and seasonal changes in butterfly wing pigmentations [17], the iterative organization of the plant body enables the plant to tailor individual developing leaves to heterogeneous environments. This property allows us to analyse how induced phenotypes shift within individual plants over time. In carnivorous plants, many species are known to produce non-trap leaves during unfavourably cool or dry seasons when the photosynthetic benefits of carnivory are likely to be low [18,19].

*Cephalotus* produces two distinct types of leaves: flat leaves for specialized photosynthesis and pitcher leaves for carnivory [11], the latter of which can also perform photosynthesis with reduced efficiency [20]. This leaf dimorphism is considered to be heterophylly because seasonal changes have been implicated in the development of this perennial species [11,20–24], and it has recently been shown to be temperature-dependent [25]. Unlike other well-studied heterophyllous plants that often produce simple and compound leaves that differ in leaf segmentation [16], *Cephalotus* shows the leaf shape alteration accompanying the so-called 'cross-zone' formation, which is associated with the change in adaxial–abaxial polarity rather than in leaf compoundness, leading to the production of pitcher- and flat-shaped leaves (figure 1a,b) [26,27]. In addition, *Cephalotus* spontaneously produces malformed leaves with apparently intermediate morphologies between flat and pitcher shapes [21,22,24,28,29]. Those malformed leaves often compromise leaf flatness, an important parameter for efficient photosynthesis [27], and tend to lack key features of functional traps, such as the lid and the vertical wall, which prevents prey from escaping (figure 1c–i). Therefore, malformed leaves appear to be non-optimal for either carnivory or photosynthesis, and can be interpreted as a consequence of a maladaptive response usually masked as a hidden reaction norm. An analysis of environmental cues that induce the formation of malformed leaves would, therefore, allow us to investigate how the hidden reaction norm is exposed.

To examine the role of covarying cues in masking the hidden reaction norm, we analysed the heterophylly of *Cephalotus* by obtaining 19 738 leaves from 3241 genetically identical plants grown under various conditions. We show that an unusual combination of temperature and photoperiod unmasks a hidden reaction norm, even though neither environmental cue is extreme.

## 2. Results

### (a) *Cephalotus* produces various forms of malformed leaves

*Cephalotus* produced malformed leaves (figure 1) at a low frequency under a constant temperature and continuous light (14%, 56/410 leaves at 15°C; 8%, 111/1,454 leaves at 25°C). Their morphology varied; some leaves were equipped with both flat-leaf-like lamina and pitcher-leaf-like features (figure 1c–f), whereas others might be better described as compromised pitcher leaves (figure 1g–h). On the basis of their morphology, the malformed leaves are unlikely to perform as well at either photosynthesis or prey capture as the flat or pitcher leaves. We classified those leaves into nine types, including regular flat and pitcher leaves, using eight morphological characters (figure 1; electronic supplementary material, figure S1 and table S1). These types are hereafter referred to using the labels illustrated in figure 1. Some of these and other types of malformed leaves have historically been reported for *Cephalotus* grown in natural habitats or cultivated conditions [11,21,22,28,29]. Although we could not exclude the possibility that our clonally propagated samples have a different genetic background than the plants the authors of previous studies observed, this observation implies that the shapes of the malformed leaves are determined in a context-dependent manner.

### (b) The heterophylly of *Cephalotus* is strictly dimorphic in response to temperature

We previously reported that flat and pitcher leaves dominated at 15°C and 25°C, respectively [25]. Consistently, a 20-week-long growth experiment showed that, along with the development of the phyllotactic spiral (figure 2a), the leaf fate was progressively segregated between 15°C and 25°C, respectively, in continuous light (figure 2b). Although too low of a temperature may induce shoot dormancy [11], the range of temperatures used here allowed plants to grow continuously with different growth rates (figure 2c). After 12 weeks, plants grown at 25°C produced ca 2.1 leaves more than those at 15°C (figure 2c).

Because this range of temperatures (15–25°C) is within that of the daily mean temperature in the natural growing season of *Cephalotus* (figure 2d; electronic supplementary material, figure S2), we explored whether the heterophylly is strictly dimorphic by examining the effect of intermediate temperatures. The patterns of phenotypic plasticity are categorized into continuous or discontinuous reaction norms. Continuous reaction norms show phenotypes changing incrementally in response to changing environmental parameters, whereas discontinuous reaction norms have thresholds in sensing

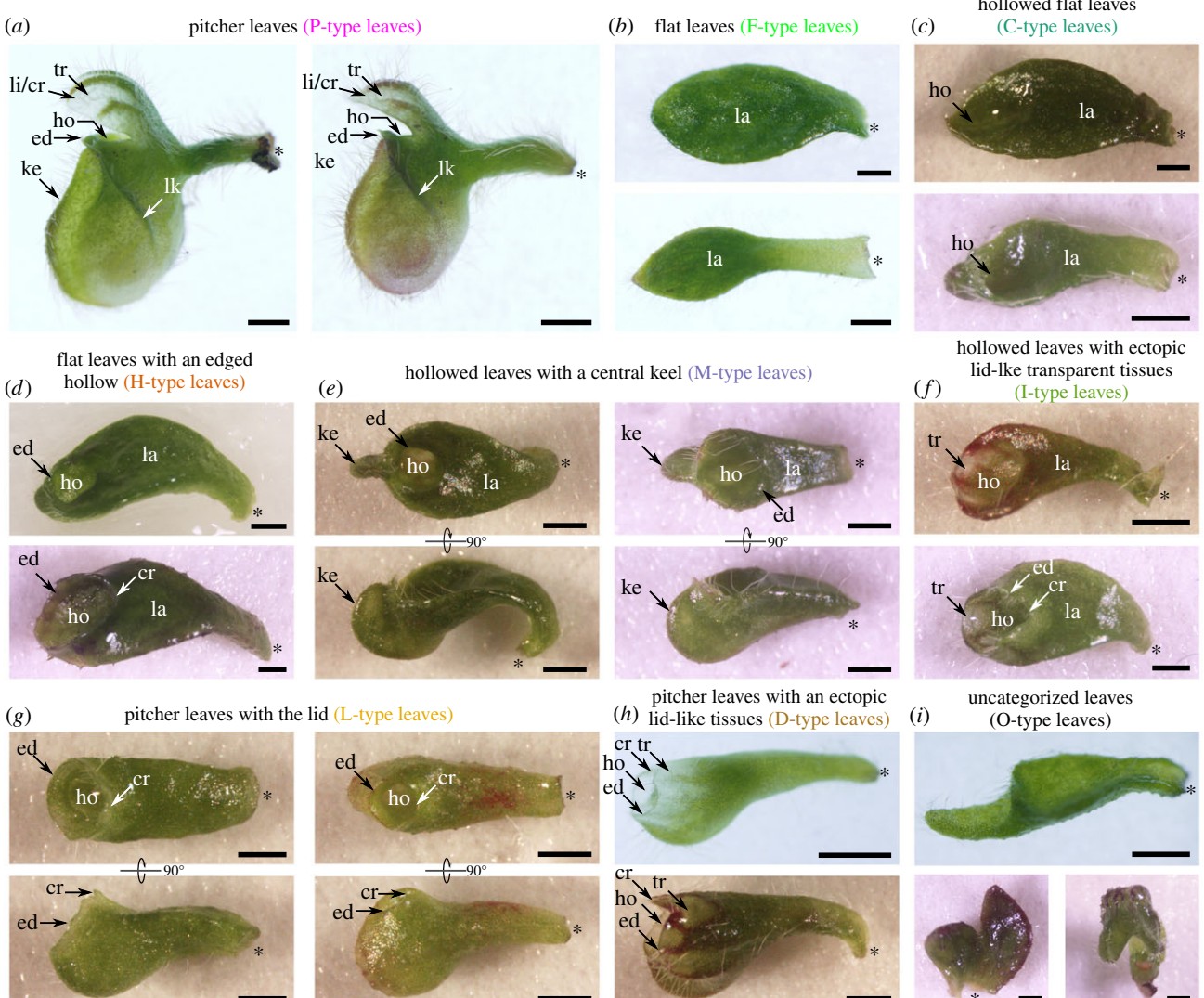

**Figure 1.** Classification of malformed leaves into eight categories. Although *Cephalotus* predominantly develops pitcher leaves (*a*) and flat leaves (*b*), malformed leaves (*c–i*) are also spontaneously produced. To show morphological variation within the categories, two or three leaves each are presented. Bars indicate 1 mm. Asterisks show the position of the leaf base. la, laminar portion; ho, hollow; ed, hollow edge; ke, central keel; lk, lateral keel; tr, lid-like transparent tissue; li, distinct lid; cr, protrusion in the cross zone. Further descriptions are provided in electronic supplementary material, figure S1. Text colours match those used in other figures. (Online version in colour.)

environments [30] (figure 2*e*). In intermediate environments, continuous reaction norms generate intermediate phenotypes. A smooth transition of multimorphic states, which we refer to here for more than two states, yields similar outcomes, even if the reaction norm is discontinuous. When discontinuous reaction norms result in dimorphic responses, intermediate environments induce the two phenotypes in a threshold-dependent manner with stochasticity.

If a *Cephalotus* plant produces leaves with an intermediate morphology more frequently at the intermediate temperatures (17.5°C, 20.0°C or 22.5°C used here) than at 15.0°C or 25.0°C, its heterophylly should not be interpreted as being dimorphic (figure 2*e*). Here, the hypothetical intermediate leaves would have a mixture of morphological characters seen in flat and pitcher leaves: i.e. a laminar portion plus one or more of the characters of pitcher leaves. Among the malformed leaves we identified, four categories fulfilled this criterion on the basis of eight characters we used (C, H, M and I; electronic supplementary material, table S1), whereas the other categories (L, D and O) require a more detailed characterization to assess their intermediateness.

We examined the leaf morphology of plants grown in growth cabinets under each of the five temperature conditions for 12 weeks (figure 2*f*). Similar proportions of the intermediate categories were produced in the intermediate temperatures (3.6%, 7.4% and 4.2%, at 17.5°C, 20.0°C and 22.5°C, respectively), and were comparable to those observed at 15.0°C and 25.0°C (12.2% and 7.4%, respectively). The result did not change when the other malformed leaf categories (L, D and O) were included. This suggests that the heterophylly of *Cephalotus* is a discontinuous reaction norm with strictly dimorphic responses, illustrating that its leaf development is robustly specialized into either flat- or pitcher-leaf morphogenesis, even at borderline temperatures.

## (c) Co-occurrence of morphological characters contributes to the dimorphic response

Excluding the two characters that structurally depend on others (hollow edge depends on the presence of a hollow, and distinct lid depends on the cross-zone formation), there are six traits that combinatorially determine the leaf categories.

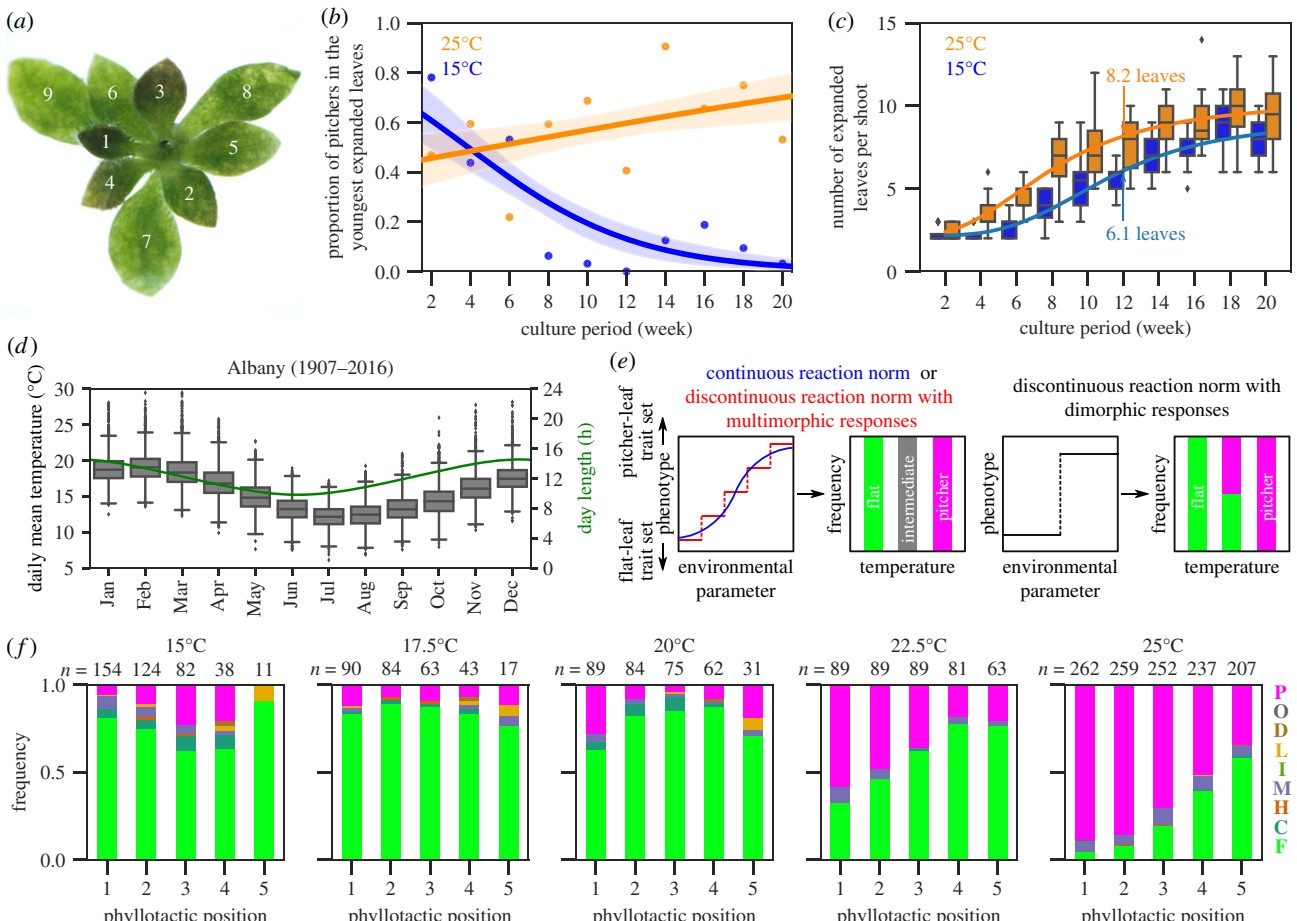

**Figure 2.** Heterophylly rarely fails in response to different temperatures. (*a*) An example of the phyllotaxis in *Cephalotus*. Expanded leaves are numbered from the youngest (1) to the oldest (9). In this plant, all expanded leaves are flat. (*b*) Logistical regression of pitcher formation rates in the youngest expanded leaves over 20 weeks. A 95% confidence interval is shaded. Each point corresponds to independent observations of 32 plants. (*c*) The leaf production rates of the main shoots at 15°C and 25°C. The curves show four-parameter logistic regressions. (*d*) Seasonal changes in temperature and photoperiod in the habitat of *Cephalotus* (Albany, Australia). Box plots describe the distribution of estimated daily mean temperatures from 1907 to 2016. The curve indicates the seasonal change in the day length. (*e*) Hypothetical outcomes of continuous and discontinuous reaction norms in the heterophylly of *Cephalotus*. We define 'intermediates' as leaves with a mixture of flat-like and pitcher-like characters. A set of analysed characters are described in the main text and summarized in electronic supplementary material, table S1. (*f*) Heterophylly at different temperatures in continuous light. The colours of stacked bar plots and corresponding leaf category labels match those in figure 1. The numbers of observed leaves (*n*) are indicated above the stacked bar plots. (Online version in colour.)

Theoretically, 64 leaf categories can arise from combinations of the six binary characters, but our observation was limited to only 21 categories (excluding category O) even when all possible character combinations were taken into account in some categories with optional characters (i.e. '+/−' in electronic supplementary material, table S1). By employing a rarefaction curve analysis of 2988 leaves produced under a constant temperature and continuous light (figure 2*f*), we first examined whether the observable leaf categories are saturated in our dataset. The curve quickly reached the near-plateau phase, representing good sample sizes (electronic supplementary material, figure S3A–B); however, the slope was still not negligible, suggesting that a substantially larger scale of leaf sampling could yield new, very rare leaf categories even under constant growth conditions because of the continuous range of developmental abnormalities.

Next, we tested whether the level of the near-plateau phase is bounded by the co-occurrence of the six characters. A null expectation with no between-character correlation was obtained by resampling. In this procedure, character states were randomly resampled from the character-wise

observed frequencies, and therefore any correlations between characters are broken while preserving the character frequencies. The resulting expectation curve had its near-plateau level at around 50 categories (electronic supplementary material, figure S3B), which was substantially higher than the number observed (ranging from 10 to 15 categories). This result illustrates a tight co-occurrence of characters, likely reflecting the robust formation of either flat or pitcher leaves, and led us to ask whether such tight co-occurrence is compromised in the malformed leaf production.

To address this question, we performed the same analysis with only the 232 malformed leaves by excluding flat and pitcher leaves. In this dataset, the gap between the observation and expectation curves was still present (electronic supplementary material, figure S3C), suggesting the character occurrence is not completely random; however, the gap between the two curves was smaller than that in all leaves. This result suggests that the tight character co-occurrence is relaxed during malformed leaf production. Taken together, these data suggest that the inter-correlated character occurrence contributes to maintaining the dimorphic response and to reducing the production of malformed leaves.

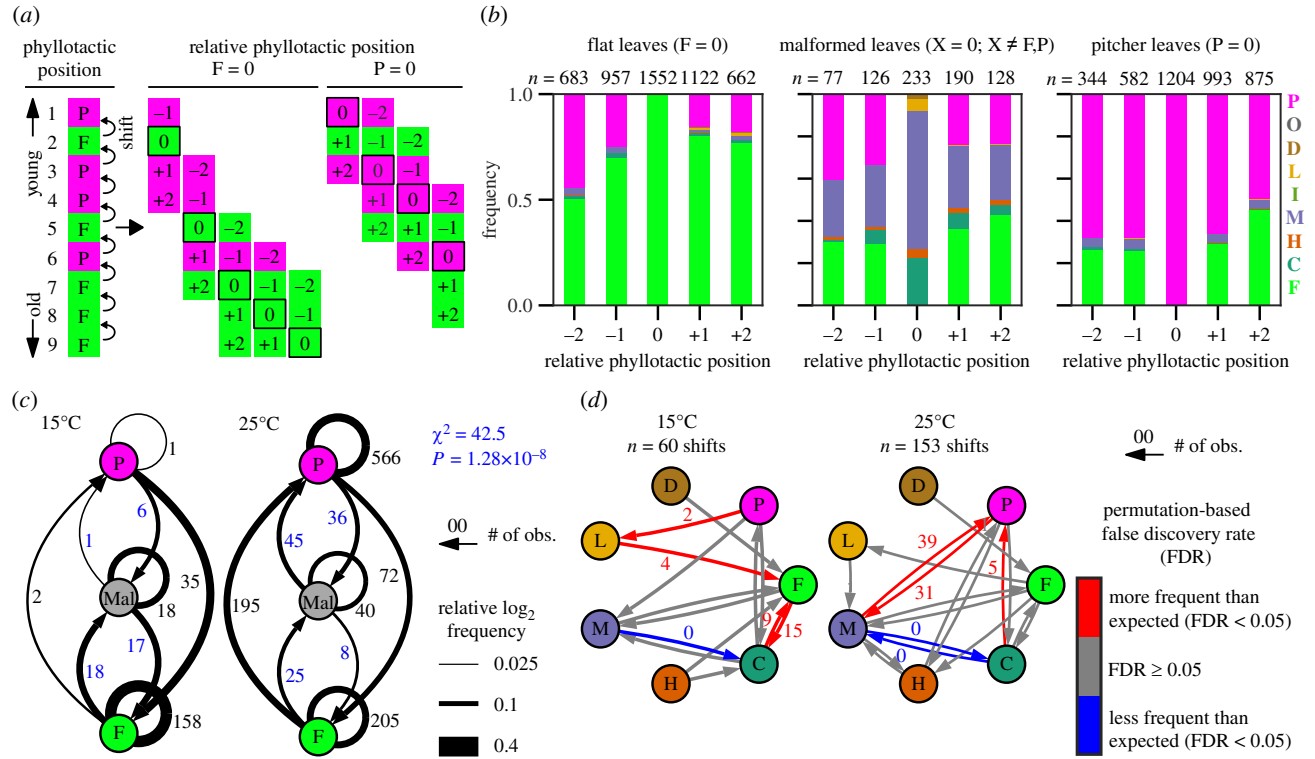

**Figure 3.** The shifts of leaf fates along the phyllotactic spiral. (*a*) Analysis of leaf fate shifts neighbouring a particular leaf category using relative phyllotactic positions. (*b*) Relative frequencies of leaf categories in the phyllotactic neighbours of flat, malformed and pitcher leaves. The leaf categories and phyllotactic positions were acquired from plants grown at 15.0°C, 17.5°C, 20.0°C, 22.5°C and 25.0°C under continuous light, and the data were pooled. The numbers of observed leaves (*n*) are indicated above the stacked bar plots. The colours of stacked bar plots and corresponding leaf category labels match those in figure 1. (*c*) Leaf fate transitions among flat, malformed and pitcher leaves. All categories of malformed leaves are amalgamated into the 'Mal' node. Observed numbers of shifts are indicated with arrows. For example, 35 on P → F means that pitcher-to-flat shifts were observed 35 times in our dataset. The arrow widths correspond to the relative proportions of log$_2$ frequency. The result of a $\chi^2$ test is shown for P ⇆ Mal and F ⇆ Mal shifts (blue). (*d*) Leaf fate transitions among individual categories. The false discovery rates (FDRs) are calculated by 10 000 permutations. Direct shifts between P and F were excluded in this analysis to highlight the transitions among malformed leaves. Category 0 is not shown because it represents multiple forms of malformed leaves. Category I is not shown because it was not included in the dataset. (Online version in colour.)

## (d) The consecutive production of malformed leaves

Multiple leaf fate determination events can occur in a single shoot tip; therefore, malformed leaf production may occur either independently or consecutively. If the malformed leaf formation is a completely stochastic event in any given leaf primordium, the malformed leaves should be produced sporadically in any phyllotactic position. To test this hypothesis, we examined the serial transitions of leaf fates during shoot development. Using the data obtained from the plants grown under constant temperature and continuous light (figure 2*f*), we aligned the phyllotactic series of leaves to analyse the leaves adjacent to particular leaf categories. In this analysis, we converted the phyllotactic positions into sequential orders relative to the position of leaves with a particular category (figure 3*a*); for example, when pitcher leaves are analysed (i.e. P = relative phyllotactic position 0), the leaves produced just before (older) and after (younger) a particular pitcher leaf are labelled +1 and −1, respectively. If malformed leaf productions are independent of each other, then flat, pitcher and malformed leaves should exhibit similar frequencies of malformed leaves in their adjacent positions. Contrary to this null expectation, our analysis showed that, compared with flat and pitcher leaves, malformed leaves had a higher probability of being adjacent to other malformed leaves ($\chi^2 = 392.5$, $p = 2.38 \times 10^{-87}$, $\chi^2$ test), illustrating their consecutive production (figure 3*b*). Because the plants were grown under constant environments, the consecutive tendency of

the malformed leaf production is best interpreted as an effect of the environmental shock at the onset of experiments. In this view, the instantaneous disturbances can influence the fate of multiple primordia by causing a long-lasting state change of the shoot apical meristem or simultaneous state changes in developing leaf primordia. Nevertheless, these results suggest that malformed leaf production is not a completely random event and possibly reflects growth conditions.

## (e) Temperature reshapes the propensity of leaf category transitions

Next, we examined if the transition tendency among leaf categories is temperature-dependent. Again, we used the data obtained from the plants grown under constant temperature and continuous light (figure 2*f*) to analyse the shifts among leaf categories along phyllotactic leaf positions (figure 3*a*). At 25°C, flat-to-malformed and malformed-to-pitcher transitions were observed more frequently than the reverse transitions (figure 3*c*). This transition tendency was reversed at 15°C, although the flat-to-malformed transitions were as abundant as the reverse direction in this dataset. This distinction of the shift propensities at 15°C and 25°C was supported by a $\chi^2$ test of the shift counts (figure 3*c*). These results suggest that malformed leaves frequently arise during the transition between flat- and pitcher-leaf-producing states. The transition analysis among individual leaf categories

showed that different categories of malformed leaves tended to mediate transitions between flat and pitcher leaves depending on temperature: C- and L-type leaves at 15°C, and C- and M-type leaves at 25°C (figure 3d). Overall, the data showed that the transition propensity was temperature-dependent, despite the stably low frequency of malformed leaf production over a range of temperatures (figure 2f).

## (f) Specific combinations of temperature and photoperiod expose the hidden reaction norm

To test the hypothesis that discordance in covarying cues can reveal hidden reaction norms, we investigated another environmental factor that influences heterophylly. We especially focused on photoperiod because it covaries with temperature in the natural habitats of Cephalotus (figure 2d), and because diverse organisms exhibit photoperiodism-related phenotypic plasticity [31]. Also, the photoperiod could be of particular importance to heterophylly in carnivorous species. The cost–benefit model of the evolution of carnivorous lifestyles predicts that abundant light enables the plant to compensate for the cost of trap leaf production by enhancing photosynthesis alongside the acquisition of prey-derived nutrients [18,19,32]. Because such a compensation cannot be accomplished in light-limited conditions, seasonal changes in light availability may explain the evolution of seasonal heterophylly in carnivorous species, including Cephalotus [18,19].

We thus evaluated the effects of short-day (8 h light : 16 h dark; 8L16D) and long-day (16L8D) conditions in combination with two temperatures (15°C and 25°C) in 12-week growth experiments (figure 4a). Although the natural growth habitats of Cephalotus do not reach such extreme day lengths (figure 2d), we chose these conditions in anticipation of clear phenotypic responses. An extremely short day length (8L16D) in combination with the summer-high temperature (25°C) frequently induced M-type malformed leaves, which was not the case when combined with the low temperature (15°C) (figure 4a). The extremely long day length (16L8D) was not associated with malformed leaf production (figure 4a).

Because a striking effect was observed in the extremely short-day photoperiod at a high temperature, we next examined similar, but more benign, conditions: 10L14D, 12L12D and 14L10D, alongside intermediate temperatures. The short-day (10L14D) and neutral (12L12D) photoperiods induced a large proportion of malformed leaves at 25°C (figure 4b). In addition, those day lengths tended to induce slightly higher proportions of malformed leaves in combination with lower temperatures (20°C and 22.5°C). A binomial generalized linear model fitted to the forms of the youngest expanded leaves (i.e. phyllotactic position = 1; figure 4a,b) indicated that the probability of malformed leaf production was significantly influenced by both temperature (coefficient = 0.346; $z = 10.6$; $p = 3.21 \times 10^{-26}$) and photoperiod (coefficient = –0.355; $z = –10.4$; $p = 3.43 \times 10^{-25}$). These observations contrast with the robust dimorphism observed under continuous lighting (figure 1f), and highlight the role of the combinatorial effects of temperature and photoperiod in exposing the hidden reaction norm.

To evaluate the effect of circadian gating [33], we examined the heterophylly under different daytime/night-time temperatures combined with the extremely short or long day length (8L16D or 16L8D). The effect was not substantial, suggesting the robustness of our results (electronic supplementary material, text S1 and figure S4).

## (g) A sparse parameter space exposes the hidden reaction norm

Although the individual cues are within the range of regular fluctuations in the habitat, the combinations of temperature and photoperiod in which we obtained many malformed leaves is rarely observed (figure 4c). An examination of the historical occurrences of the summer-high daily mean temperatures (greater than 25°C) and the short-to-neutral day lengths (less than 12 h) showed that the conditions resulting in malformed leaves have individually happened in Cephalotus habitats on a yearly basis (figure 4d). By contrast, their combinations were quite rare (figure 4d), totalling only 6 days in 67 years at Albany (from 1907 to 2016, excluding the years with greater than 30 days of missing records). Cephalotus is a perennial herb that reaches its maturity 4–6 years after germination, and whose generation length is estimated to be approximately 10 years [23,34]; therefore, such variable combinations are, on average, a once-in-a-lifetime event or rarer. The heterophyllous trait in Cephalotus may not have been sufficiently exposed to such parameter spaces for natural selection to work.

Cephalotus produced high proportions of malformed leaves under the short-to-neutral-day/higher-temperature conditions and, based on the parameter distributions (figure 4c), it is possible that the opposite conditions (long-day/lower-temperature) could also induce a failed plasticity. Growth experiments at a lower temperature (e.g. 10°C) were not successful because of shoot dormancy, however, and we therefore focused on the high-temperature/short-to-neutral-day conditions.

## (h) Parameter combinations that generate higher-order sparsity

The frequency of the malformed-leaf-inducing conditions (6 days in 67 years at Albany, Australia) is one order of magnitude smaller than the expected occurrence (54 days in 67 years) calculated from the individual probabilities (figure 4d). To better understand how sparse parameter spaces are generated, we performed a numerical simulation. Day length and daily mean temperature (histograms in figure 4c) were modelled using a beta distribution ($\alpha = 0.583$; $\beta = 0.590$; location = 9.81; scale = 4.71) and a Gaussian distribution (location = 15.2; scale = 3.51), respectively. By linking them with the observed correlation (Pearson's $r = 0.61$), simulated data were generated. Even with a simulation corresponding to a long period of time (36 500 data points, or 100 years), the simulations failed to fill parameter spaces that correspond to the observed sparse parameter spaces in temperature and day length (electronic supplementary material, figure S5). By modifying the simulation model, we established that such sparsity did not persist without the environmental cue covariation (electronic supplementary material, figure S5). The shape of the parameter distribution appears to be another important factor because its manipulation changed the presence and shape of the sparse parameter spaces; for example, the combinations of two cues that follow the photoperiod-like beta distribution did not expose a sparse parameter space even though the correlation was maintained, whereas a pair of variables following the temperature-like Gaussian distribution expanded the area of sparse parameter spaces (electronic supplementary material, figure S5). These results highlight the importance of environmental cue variations

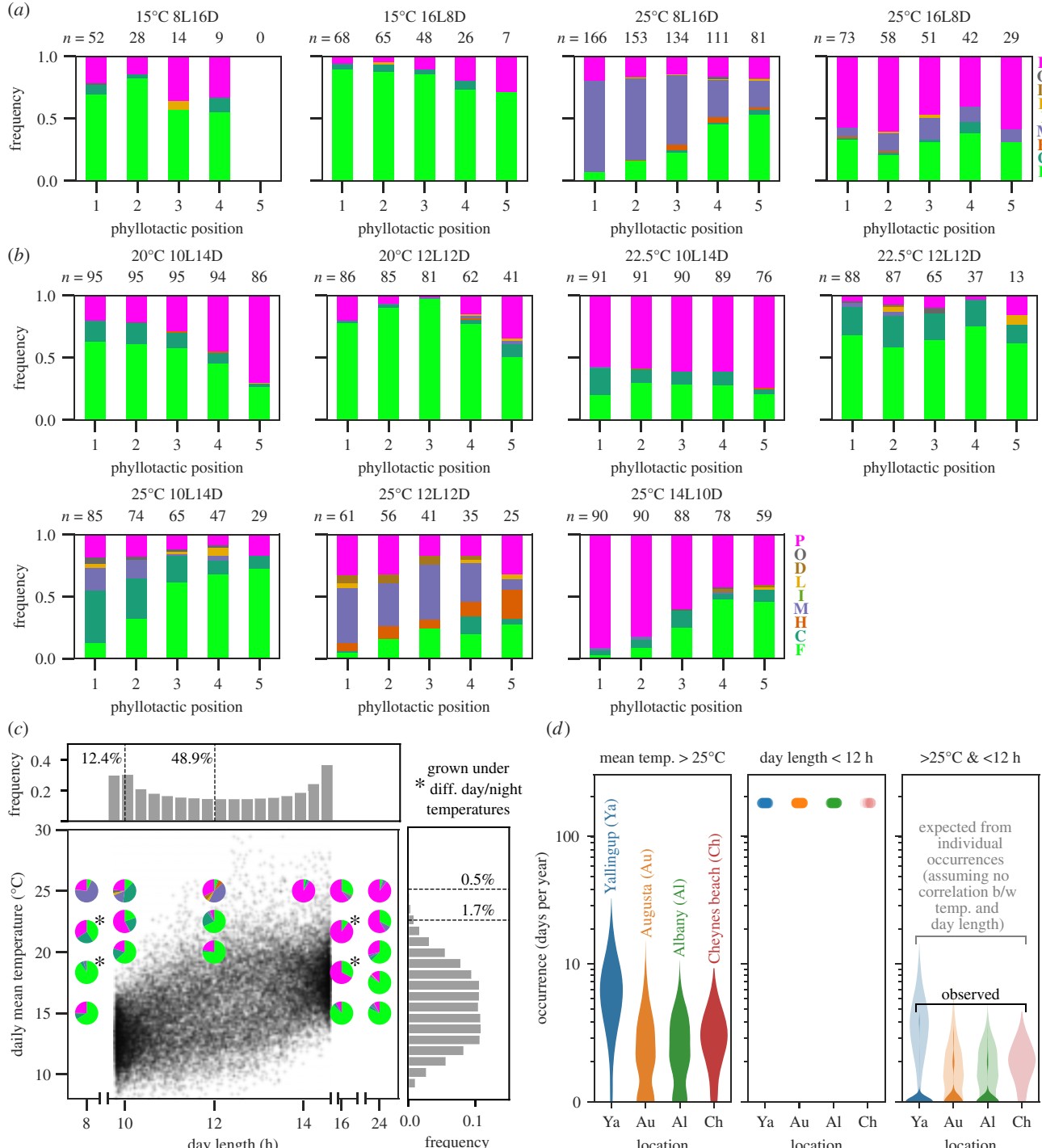

**Figure 4.** Heterophylly changes in discordant combinations of temperature and photoperiod. (*a,b*) The effect of different combinations of temperature and photoperiod. Both extreme (*a*) and benign (*b*) day lengths were examined in combination with various temperatures. The numbers of observed leaves (*n*) are indicated above the stacked bar plots. The colours of stacked bar plots and corresponding leaf category labels match those in figure 1. (*c*) The rare occurrence of the malformed-leaf-inducing combinations of environmental conditions in the natural *Cephalotus* habitats. The scatter plot and histograms show the relationships between day length and estimated daily mean temperature from 1907 to 2016 in Albany, Australia. Points correspond to days. Pie charts indicate the relative frequency of leaf categories at the youngest expanded leaf positions we observed under the controlled environments (figures 2 and 4; electronic supplementary material, figure S4). Asterisks indicate the results with different daytime and night-time temperatures (electronic supplementary material, figure S4). The colours indicate leaf categories in *a–b*. (*d*) Historical yearly occurrences of summer-high daily mean temperature (greater than 25°C, left), short-to-neutral day length (less than 12 h, middle) and their combinations (right). Meteorological data from four localities (electronic supplementary material, figure S2) are shown. (Online version in colour.)

and covariations in shaping the sparsity of higher-order parameter spaces.

Taken together, our experiments and simulation reveal a potential role of environmental cue combinations in the heterophylly of *Cephalotus*. Our results provide a new perspective that a discordant combination of covarying factors can expose hidden reaction norms even in a benign range of individual environmental factors.

## 3. Discussion

In this study, we showed that naturally rare combinations of multiple environmental cues expose a hidden reaction norm (figure 4), which is otherwise masked (figures 2–4; electronic supplementary material, figure S4). *Cephalotus* produced diverse shapes of malformed leaves (figure 1) that are unlikely to perform as well at either photosynthesis or prey

capture as the flat or pitcher leaves, suggesting a maladaptive response. It is possible that the populations of *Cephalotus* would purge the genetic components underlying the hidden reaction norm if such an unfavourable property is manifested; however, there are two possibilities to explain why *Cephalotus* has maintained this plasticity. One possible mechanism is the developmental bias [35], which can prevent the elimination of malformed leaf production even under strong natural selection because of the mutational inaccessibility to developmental solutions. The other possibility is that the variable combinations were so rare that the maladaptive responses could not be efficiently exposed to the act of natural selection. Although these explanations are not mutually exclusive, the historically limited occurrence of the malformed-leaf-inducing combinations of temperature and photoperiod (figure 4c,d) supports the latter idea. In this view, malformed leaf production happens infrequently and is therefore only slightly deleterious. The resultant selection coefficient may not be large enough to purge the slightly deleterious trait in the relatively small population sizes of *Cephalotus*; only approximately 4000 mature individuals exist across all populations, which generally contain fewer than 100 plants each according to the latest IUCN Red List of Threatened Species [34].

Our data on the heterophylly of *Cephalotus* raise a prediction that adaptive phenotypic plasticity could be compromised not only by extreme cues, such as a heat stress that increases mortality rate if it persists [36], but also by unprecedented combinations of benign cues in the N-dimensional parameter spaces of complex environments. In line with this idea, an *A. thaliana* relative (*A. halleri* subsp. *gemmifera*) shows decreased fitness and a disturbed transcriptome when exposed to unusual combinations of temperature and photoperiod [37]. The roles of daily fluctuations and microhabitat differences (e.g. ambient versus soil temperatures) are of interest for future study because it is possible that they may readjust the reaction norm to mask the less-adaptive phenotypes in the face of temperature–photoperiod discordance.

Uncommon combinations of environmental cues other than temperature and photoperiod may also expose hidden reaction norms (see electronic supplementary material, text S2 for further discussion). Indeed, sparse parameter spaces are prevalent among popularly recorded environmental variables (electronic supplementary material, figure S6), such as precipitation, solar radiation, atmospheric pressure, humidity and wind speed. Because plant genomes harbour a certain number of genes whose expression is better correlated to one of those variables than the others [38], their discordance may result in hidden phenotypes through abnormal gene expression profiles. Thus, the analysis of other factors as well as other species will provide further insights into the generality of our findings on the hidden reaction norm. Consistent with our simulation showing that the generation of higher-order sparsity is conditional to individual cue distributions and their correlation (electronic supplementary material, figure S5), certain variable combinations did not show the higher-order sparsity (electronic supplementary material, figure S6; e.g. day length and wind speed). The choice of environmental variable combinations to test will, therefore, be an important consideration for future study.

Historical records of air temperature show an upward trend in the habitats of *Cephalotus* (electronic supplementary material, figure S7), meaning that the changing climate tends to increase the occurrence of previously infrequent high-temperature/ short-to-neutral-day combinations. Organisms with a very limited range of distribution, such as *Cephalotus* [11] (electronic supplementary material, figure S2), are predicted to have difficulty in responding to climate change through migration, and their evolutionary rescue in their current habitats due to adaptive evolution is considered to be rare [39]. Indeed, only 26 out of the 114 historically recorded *Cephalotus* populations are confirmed to exist at present [11]. Future studies should address how organisms overcome discrepant cues in the face of current climate change, which alters the covariance structure of environmental variables [10].

## 4. Material and methods

Experimental details are available as electronic supplementary material, Methods.

Data accessibility. Tabulated data, package versions and code to reproduce our results are available in electronic supplementary material, table S2 and Data.

Authors' contributions. K.F. and M.H. conceived the study and wrote the paper. K.F., H.N. and G.P. conducted the experiments. K.F. analysed the data and performed the numerical simulations. M.H. supervised the project. All authors commented on the paper.

Competing interests. We declare we have no competing interests.

Funding. We acknowledge the following sources of funding: MEXT/ JSPS KAKENHI grant no. 12J04926 (K.F.), 22128001 (M.H.), 22128002 (M.H.) and 17H06390 (M.H.), and the Sofja Kovalevskaja programme from the Alexander von Humboldt Foundation (K.F.).

Acknowledgements. We thank Carl Schlichting for valuable comments. This work was supported by MPRF-NIBB for cultivation and Yoko Matsuzaki and Hatsumi Fukada for the experiments.

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
