## [Reviewer comments · Proceedings of the Royal Society B: Biological Sciences]

Review History

RSPB-2020-1546.R0 (Original submission)

Review form: Reviewer 1 (Thomas Givnish)

Recommendation

Accept with minor revision (please list in comments)

Scientific importance: Is the manuscript an original and important contribution to its field?

Excellent

General interest: Is the paper of sufficient general interest?

Excellent

Quality of the paper: Is the overall quality of the paper suitable?

Excellent

Is the length of the paper justified?

Yes

Should the paper be seen by a specialist statistical reviewer?

No

Do you have any concerns about statistical analyses in this paper? If so, please specify them explicitly in your report.

No

It is a condition of publication that authors make their supporting data, code and materials available - either as supplementary material or hosted in an external repository. Please rate, if applicable, the supporting data on the following criteria.

Is it accessible?

Yes

Is it clear?

Yes

Is it adequate?

Yes

Do you have any ethical concerns with this paper?

No

Comments to the Author

Fukushima et al. demonstrate that the heterophyllous Western Australian pitcher plant (*Cephalotus follicularis*) exhibits a maladaptive reaction norm when exposed to combinations of growth temperature and daylength that occur rarely in the wild. Their meticulous analysis shows that a clear switch from flat non-trap leaves to cylindrical trap leaves dependent on temperature, but a range of abnormal leaf phenotypes when the conditioning temperature and daylength are in combinations seen only rarely – in fact, once in a lifetime, based on the authors' analysis of climatic data from part of the range of *Cephalotus*. This is a brilliant paper, and should be of interest to ecologists, evolutionary biologists, and developmental biologists.

I had three minor suggestions that the author should consider in revising their manuscript:

1. The statement on lines 107-108 ("There was no indication that their morphology was more suitable for photosynthesis or carnivory than flat or pitcher leaves, respectively.") and its echo in the Discussion are rather backhanded. I appreciate the authors' careful attempt to not claim what they have not demonstrated. But I do wonder whether a more positive statement – something along the lines that intermediate leaves are unlikely to perform as well at either photosynthesis or prey capture as the naturally occurring flat or trap leaves – might convey their conclusion more clearly.
2. The authors might want to cite Givnish et al. 1984 (already in the bibliography) or Givnish et al. 2018 in Ellison and Adamec) explaining that some carnivorous plants produce non-trap leaves during unfavorable cool or dry seasons when prey density or the photosynthetic benefits of carnivory are likely to be low. In the case of *Cephalotus* inhabiting the (permanently wet) reed swamps near Albany, summer comes with greater prey density and light availability, which tends to explain the production of trap leaves under those conditions, and the production of non-trap leaves under the cooler, cloudier, shorter days of winter.
3. The authors might also want to read Givnish 2002 (*Evolutionary Ecology*) on how the breakdown of a regional correlation of environmental cues that direct plasticity in a given set of traits might impose a deep, unavoidable limit on the extent to which selection can drive an expansion in species range. In the winter rainfall/Mediterranean climate region of SW Western Australia, variation in daylength vs. rainfall and temperature are more or less perfectly inversely and directly related, respectively. The developmental program elucidated by the authors would tend to behave in an adaptive sense within this region. But as one moves outside that region (or

simply outside the reed swamp topographic habitat that provides abundant water year-round), the correlations among the environmental signals – which also help determine the net returns from carnivory – break down. This could help limit the range of *Cephalotus* to the SW corner of Western Australia, where the regional correlation of environmental signals remains intact and produces adaptive shifts in leaf morphology in the reed swamp habitat.

Review form: Reviewer 2

Recommendation

Reject – article is scientifically unsound

Scientific importance: Is the manuscript an original and important contribution to its field?

Good

General interest: Is the paper of sufficient general interest?

Good

Quality of the paper: Is the overall quality of the paper suitable?

Poor

Is the length of the paper justified?

No

Should the paper be seen by a specialist statistical reviewer?

No

Do you have any concerns about statistical analyses in this paper? If so, please specify them explicitly in your report.

No

It is a condition of publication that authors make their supporting data, code and materials available - either as supplementary material or hosted in an external repository. Please rate, if applicable, the supporting data on the following criteria.

Is it accessible?

No

Is it clear?

No

Is it adequate?

No

Do you have any ethical concerns with this paper?

No

Comments to the Author

Review of A discordance of seasonally covarying cues unmasks a hidden reaction norm in the heterophyllous pitcher plant *Cephalotus follicularis*

I started reading this manuscript with great interest. The study system is fascinating, and the idea that novel environments may induce the production of “strange” or malformed phenotypes even when these novel environments are combinations of factors that the plants experience in the field

is really intriguing.

However, I had a serious problem with the way the manuscript is laid out (I find the contents of the different sections very disproportionate), and importantly, with the methodological description of what actually was done. The methods section in the main text is reduced to one short paragraph. I understand that not all details may fit into the main text, but this is extremely short and lacks all details. I've checked other manuscripts in the journal, and it does not seem to be a journal limitation. Even so, this would be OK if the supplementary methods were thorough. However, I also found this material confusing and incomplete. First, there is no explanation of the genetic material that was used. There is only a short sentence stating that the plants were obtained in a nursery. However, the introduction states that almost 4000 genetically-identical individuals were used. This is very confusing. Are they all clones? What is the origin of this material? Do they come from the same populations? How can the reader know if the experimental treatments are "normal" for these plants if we don't know their origin? Also, looking at the map of the distribution of the species, it is likely that different populations experience different environmental conditions, and hence, different combinations of abiotic factors. Then, justifying the origin of the material and the climatic conditions of the study material is critical. My second concern is the lack of clarity in the explanation of the experimental treatments and the experiments themselves. Is this just one experiment where light and temperature were manipulated? Is it different experiments? I've looked through Table S1, where the experiments are supposedly detailed, and I couldn't make sense of it. Furthermore, I was not find any information on how the experiments were actually performed. Were they done in a greenhouse? On growth chambers? How long did they last? Were plants potted individually? I'm just very surprised by the lack of detail of the methods section, and puzzled about how this could be at all reproducible.

Unfortunately, the lack of clarity on the experiments makes it impossible to put the results in perspective, and prevents from drawing meaningful conditions. I'm sorry I cannot be more positive.

Line 28: what do you mean by "normal"?

L30: does a hidden norm of reaction always involve the mis-regulation of phenotypes?

I would like to see some more specific methodological details included in the abstract. It should be a stand-alone document, and it is current state, it is missing key information.

L50-51: is all plasticity driven by genetic assimilation and accommodation?

L97-98: I find this phrasing odd. Plasticity is a property (the slope) of the reaction norm. What do you mean by "a hidden reaction norm in phenotypic plasticity"?

L376: do you mean there was no dark period?

Supplementary methods

A description of the species and the type of habitats where it occurs would be desirable.

L666: what is axenic culture? Please describe.

L667: where do these plants come from? Are they from the same provenance?

L681-683: not clear what this sentence adds. Please explain.

L685: what do you mean by benign?

L668-687: the experiments need to be described clearer and provide much more detail. I've looked through Table S1 and it is not clear if this was just one experiment or several experiments. Was there a first experiment comparing short vs long day (L684) at a single temperature? And then a second experiment with diverse day lengths? How were these replicates (1-5) done? With the same individuals? Where were the experiments done? In a greenhouse, growth chamber? Were plants potted individually? And most importantly, what is the genetic structure of the sample? Do all individuals come from the same genetic origin? Are they clones? The end of the introduction mentions genetically-identical individuals, but I can't find a proper description of the structure of the experimental plants.

L694: but what experiments are these? Why was this categorization used for one or other experiments?

L699: what do you mean by this? are you referring to the center of the distribution? Is it where

the species is most abundant?

Figure 1: I'd suggest numbering the panels instead of using letters. Because the leaves are categorized also by letters, it's just confusing.

Decision letter (RSPB-2020-1546.R0)

18-Sep-2020

Dear Dr FUKUSHIMA:

I am writing to inform you that your manuscript RSPB-2020-1546 entitled "A discordance of seasonally covarying cues unmasks a hidden reaction norm in the heterophyllous pitcher plant *Cephalotus follicularis*" has, in its current form, been rejected for publication in Proceedings B.

This action has been taken on the advice of referees, who have recommended that substantial revisions are necessary. With this in mind we would be happy to consider a resubmission, provided the comments of the referees are fully addressed. However please note that this is not a provisional acceptance.

Sincerely,
Dr Sasha Dall
<mailto:proceedingsb@royalsociety.org>

Associate Editor
Board Member: 1
Comments to Author:
Dear Authors

Thank you for sending your manuscript to Proc Roy Soc B for consideration. Both reviewers really enjoyed many aspects of this manuscript, although in the end, the recommendations were

conflicting. I found the manuscript quite fascinating, with obvious large implications for how organisms may adapt to climate change. Everyone who has read this manuscript can see that it has obvious potential. However, one of the reviewers was particularly scathing about the lack of information given in the methods, which makes it impossible to truly judge the scientific merits of this manuscript. I am sorry that I am unable to offer you more positive news on this manuscript but I do agree that the criticisms are valid and hence I cannot recommend acceptance.

Reviewer(s)' Comments to Author:

Referee: 1

Comments to the Author(s)

Fukushima et al. demonstrate that the heterophyllous Western Australian pitcher plant (*Cephalotus follicularis*) exhibits a maladaptive reaction norm when exposed to combinations of growth temperature and daylength that occur rarely in the wild. Their meticulous analysis shows that a clear switch from flat non-trap leaves to cylindrical trap leaves dependent on temperature, but a range of abnormal leaf phenotypes when the conditioning temperature and daylength are in combinations seen only rarely – in fact, once in a lifetime, based on the authors' analysis of climatic data from part of the range of *Cephalotus*. This is a brilliant paper, and should be of interest to ecologists, evolutionary biologists, and developmental biologists.

I had three minor suggestions that the author should consider in revising their manuscript:

1. The statement on lines 107-108 ("There was no indication that their morphology was more suitable for photosynthesis or carnivory than flat or pitcher leaves, respectively.") and its echo in the Discussion are rather backhanded. I appreciate the authors' careful attempt to not claim what they have not demonstrated. But I do wonder whether a more positive statement – something along the lines that intermediate leaves are unlikely to perform as well at either photosynthesis or prey capture as the naturally occurring flat or trap leaves – might convey their conclusion more clearly.
2. The authors might want to cite Givnish et al. 1984 (already in the bibliography) or Givnish et al. 2018 in Ellison and Adamec) explaining that some carnivorous plants produce non-trap leaves during unfavorable cool or dry seasons when prey density or the photosynthetic benefits of carnivory are likely to be low. In the case of *Cephalotus* inhabiting the (permanently wet) reed swamps near Albany, summer comes with greater prey density and light availability, which tends to explain the production of trap leaves under those conditions, and the production of non-trap leaves under the cooler, cloudier, shorter days of winter.
3. The authors might also want to read Givnish 2002 (*Evolutionary Ecology*) on how the breakdown of a regional correlation of environmental cues that direct plasticity in a given set of traits might impose a deep, unavoidable limit on the extent to which selection can drive an expansion in species range. In the winter rainfall/Mediterranean climate region of SW Western Australia, variation in daylength vs. rainfall and temperature are more or less perfectly inversely and directly related, respectively. The developmental program elucidated by the authors would tend to behave in an adaptive sense within this region. But as one moves outside that region (or simply outside the reed swamp topographic habitat that provides abundant water year-round), the correlations among the environmental signals – which also help determine the net returns from carnivory – break down. This could help limit the range of *Cephalotus* to the SW corner of Western Australia, where the regional correlation of environmental signals remains intact and produces adaptive shifts in leaf morphology in the reed swamp habitat.

Referee: 2

Comments to the Author(s)

Review of A discordance of seasonally covarying cues unmasks a hidden reaction norm in the heterophyllous pitcher plant *Cephalotus follicularis*

I started reading this manuscript with great interest. The study system is fascinating, and the idea that novel environments may induce the production of “strange” or malformed phenotypes even when these novel environments are combinations of factors that the plants experience in the field is really intriguing.

However, I had a serious problem with the way the manuscript is laid out (I find the contents of the different sections very disproportionate), and importantly, with the methodological description of what actually was done. The methods section in the main text is reduced to one short paragraph. I understand that not all details may fit into the main text, but this is extremely short and lacks all details. I’ve checked other manuscripts in the journal, and it does not seem to be a journal limitation. Even so, this would be OK if the supplementary methods were thorough. However, I also found this material confusing and incomplete. First, there is no explanation of the genetic material that was used. There is only a short sentence stating that the plants were obtained in a nursery. However, the introduction states that almost 4000 genetically-identical individuals were used. This is very confusing. Are they all clones? What is the origin of this material? Do they come from the same populations? How can the reader know if the experimental treatments are “normal” for these plants if we don’t know their origin? Also, looking at the map of the distribution of the species, it is likely that different populations experience different environmental conditions, and hence, different combinations of abiotic factors. Then, justifying the origin of the material and the climatic conditions of the study material is critical. My second concern is the lack of clarity in the explanation of the experimental treatments and the experiments themselves. Is this just one experiment where light and temperature were manipulated? Is it different experiments? I’ve looked through Table S1, where the experiments are supposedly detailed, and I couldn’t make sense of it. Furthermore, I was not find any information on how the experiments were actually performed. Were they done in a greenhouse? On growth chambers? How long did they last? Were plants potted individually? I’m just very surprised by the lack of detail of the methods section, and puzzled about how this could be at all reproducible.

Unfortunately, the lack of clarity on the experiments makes it impossible to put the results in perspective, and prevents from drawing meaningful conditions. I’m sorry I cannot be more positive.

Line 28: what do you mean by “normal”?

L30: does a hidden norm of reaction always involve the mis-regulation of phenotypes?

I would like to see some more specific methodological details included in the abstract. It should be a stand-alone document, and it is current state, it is missing key information.

L50-51: is all plasticity driven by genetic assimilation and accommodation?

L97-98: I find this phrasing odd. Plasticity is a property (the slope) of the reaction norm. What do you mean by “a hidden reaction norm in phenotypic plasticity”?

L376: do you mean there was no dark period?

Supplementary methods

A description of the species and the type of habitats where it occurs would be desirable.

L666: what is axenic culture? Please describe.

L667: where do these plants come from? Are they from the same provenance?

L681-683: not clear what this sentence adds. Please explain.

L685: what do you mean by benign?

L668-687: the experiments need to be described clearer and provide much more detail. I’ve looked through Table S1 and it is not clear if this was just one experiment or several experiments. Was there a first experiment comparing short vs long day (L684) at a single temperature? And then a second experiment with diverse day lengths? How were these replicates (1-5) done? With the same individuals? Where were the experiments done? In a greenhouse, growth chamber? Were plants potted individually? And most importantly, what is the genetic structure of the sample? Do all individuals come from the same genetic origin? Are they clones? The end of the introduction mentions genetically-identical individuals, but I can’t find a proper description of the structure of the experimental plants.

L694: but what experiments are these? Why was this categorization used for one or other experiments?

L699: what do you mean by this? are you referring to the center of the distribution? Is it where the species is most abundant?

Figure 1: I'd suggest numbering the panels instead of using letters. Because the leaves are categorized also by letters, it's just confusing.

Author's Response to Decision Letter for (RSPB-2020-1546.R0)

See Appendix A.

RSPB-2020-2568.R0

Review form: Reviewer 2

Recommendation

Accept with minor revision (please list in comments)

Scientific importance: Is the manuscript an original and important contribution to its field?

Good

General interest: Is the paper of sufficient general interest?

Good

Quality of the paper: Is the overall quality of the paper suitable?

Acceptable

Is the length of the paper justified?

Yes

Should the paper be seen by a specialist statistical reviewer?

No

Do you have any concerns about statistical analyses in this paper? If so, please specify them explicitly in your report.

No

It is a condition of publication that authors make their supporting data, code and materials available - either as supplementary material or hosted in an external repository. Please rate, if applicable, the supporting data on the following criteria.

Is it accessible?

Yes

Is it clear?

No

Is it adequate?

Yes

Do you have any ethical concerns with this paper?

No

Comments to the Author

I appreciate the author's efforts to improve the clarity of the methods. I find this revision much more improved, and it is now possible to make sense of the experiments that were done and the interpretation of the results.

In all honesty I have to say that I find some of the methodological choices of the authors somewhat problematic (e.g. the unknown origin of the material), and some of the methodological aspects of the experiment were still unclear. For instance, I could not find the light levels on the growth chamber, which usually are on the low side. I wonder how this might have affected the combination of light and temperature and whether these combinations were "normal" or not. I nevertheless find the results interesting, and leave to the editor whether the revision meets the standard for publication.

Decision letter (RSPB-2020-2568.R0)

27-Nov-2020

Dear Dr FUKUSHIMA:

Your manuscript has now been peer reviewed and the reviews have been assessed by an Associate Editor. The reviewers' comments (not including confidential comments to the Editor) and the comments from the Associate Editor are included at the end of this email for your reference. As you will see, the reviewers and the Editors have raised some concerns with your manuscript and we would like to invite you to revise your manuscript to address them.

Research ethics:

Use of animals and field studies:

It is a condition of publication that you make available the data and research materials supporting the results in the article (<https://royalsociety.org/journals/authors/author-guidelines/#data>). Datasets should be deposited in an appropriate publicly available repository and details of the associated accession number, link or DOI to the datasets must be included in the Data Accessibility section of the article (<https://royalsociety.org/journals/ethics-policies/data-sharing-mining/>). Reference(s) to datasets should also be included in the reference list of the article with DOIs (where available).

Please submit a copy of your revised paper within three weeks. If we do not hear from you within this time your manuscript will be rejected. If you are unable to meet this deadline please let us know as soon as possible, as we may be able to grant a short extension.

Best wishes,

Dr Sasha Dall
 mailto: proceedingsb@royalsociety.org

Associate Editor
 Comments to Author:

This manuscript was reviewed by 2 reviewers: one reviewer was extremely positive, while the other was positive but quite critical. We sent the manuscript back to the authors for revision and from there it was send back only to the critical reviewer. This reviewer has now taken another look at the manuscript and while positive and praiseworthy about the results and changes made, the reviewer has suggested that some of the methodological aspects are still “problematic.”

Unfortunately, the reviewer is quite vague in this regard and does not go into specifics. Only two specific concerns have been raised and I would like the authors to address both of these issues:

3) The reviewer states that the unknown origin of the plant material is problematic. While I think this is not ideal, I do not think that this problem is enough to sink the paper. Perhaps I can ask how the authors are sure that they are working with *Cephalotes follicularis*. Were there any genetic studies which conformed this identification, or is this species easily identified using morphology? Perhaps some details could be added in text.

4) The reviewer was also unable to find the light intensity levels used in the experiment. Clearly light intensity (in addition to day length and temperature), may also affect the expression of these hidden reaction norms and so it is pertinent to provide these kinds of information and state whether the intensity level fall within the norms experienced by this species

In general, I think this is a really interesting manuscript, with clear value for understanding the triggers and clues controlling phenotypically plastic traits. I think that the authors have made an excellent effort to address the first round of comments and I found the science compelling and detailed. In addition to the comments made by the reviewer, I also read through the manuscript and below, I suggest various comments and edits

Title: I find the title a little hard to digest and I only realize what it means after reading the entire manuscript. Is it possible to make this title easier to understand for people who are not so familiar with this field? I think that part of the problem concerns the terminology “unmasks hidden reaction norm” which to me is rather cryptic as before reading this manuscript I had no idea what a hidden reaction norm was.

L52: not sure exactly what is meant by “genetic assimilation and accommodation”

L64: I think that clarity may be improved by adding a word about autocorrelation (your decision): Under conditions typical of a habitat, highly auto-correlated environmental cues often change concurrently, as observed in the seasonal perturbations of temperature and photoperiod.

L66: I have suggested changes to this sentence for clarity and impact. Please ensure that this is what you meant if you end up accepting it: However, here we test the idea that when usually auto-correlated environmental cues become decoupled (as one may expect under climate change scenarios), normally hidden reaction norms may become exposed.

L71: Due to their low motility, phenotypic plasticity is likely an especially important adaptive trait in plants, which regularly experience environments that are heterogeneous in space and time. Frequently, such plasticity is observed as variation in leaf forms and shapes where it is termed heterophylly.

L90: Unlike other well-studied heterophyllous plants, *Cephalotus* shows the leaf shape alteration accompanying the so-called ‘cross-zone’ formation, which is associated with the change in adaxial-abaxial polarity leading to pitcher development [26,27]. I have no idea what you are trying to say here – please clarify by rewriting.

L95: Are there any refs to support that these leaves are not optimal for either function. This is an important assumption of this manuscript. Refer to the relevant parts of figure 1 for graphical support

Supplementary methods:

L78: You write: "Cephalotus, which is currently listed as a "vulnerable" species" But cephalotus is a genus, so you either need to change species to genus or you need to add in a species name after Cephalotus

L78: Does this mean that all plant material is effectively clonal with little to no genetic variation? If so, write it.

L92: I think you need to describe what you mean by adult and juvenile pitcher leaves

L93: what does heteroblastic mean? Please try not to use jargon. Or at the very least, explain the terminology

L96: I am not 100% sure that I understand what you mean by "phenotypic plasticity between flat leaves and juvenile pitcher leaves in Cephalotus" No you mean that the hidden reaction norms exposed by this expt are phenotypes which are intermediate, between flat and juvenile leaves. It may be a good idea to show photographs of the various kinds of leaves. Perhaps refer to figure 1.

L181: Not sure if the ref to S2-S2 is correct. It seems like you want to refer to a number of sup figs here, but S2-S2 does not make sense

Figures.

Not clear what the color coding means in 3B and D. Similar comments for Fig 4 A-C. I think that the leaf color categories refer to Fig 1 (but this is not clear in all of the figure legends).

Furthermore, the colors in Fig 1 are not exactly the same as the colors in figs 3 & 4, making it hard to move between figures. I think that you could use exactly the same colors in the figures by making thick borders differing in color around each of the pictures in Fig 1.

Reviewer(s)' Comments to Author:

Referee: 2

Comments to the Author(s).

I appreciate the author's efforts to improve the clarity of the methods. I find this revision much more improved, and it is now possible to make sense of the experiments that were done and the interpretation of the results.

In all honesty I have to say that I find some of the methodological choices of the authors somewhat problematic (e.g. the unknow origin of the material), and some of the methodological aspects of the experiment were still unclear. For instance, I could not find the light levels on the growth chamber, which usually are on the low side. I wonder how this might have affected the combination of light and temperature and whether these combinations were "normal" or not. I nevertheless find the results interesting, and leave to the editor whether the revision meets the standard for publication.

Author's Response to Decision Letter for (RSPB-2020-2568.R0)

See Appendix B.

Decision letter (RSPB-2020-2568.R1)

29-Dec-2020

Dear Dr Fukushima

I am pleased to inform you that your manuscript entitled "A discordance of seasonally covarying cues uncovers misregulated phenotypes in the heterophyllous pitcher plant *Cephalotus follicularis*" has been accepted for publication in Proceedings B.

Open Access

Paper charges

Sincerely,

Dr Sasha Dall

Associate Editor:

Board Member

Comments to Author:

Thank you for making such a good effort at addressing at addressing my comments and the comments from one of the reviewers. I think that the manuscript is now much stronger than the original version and it is likely to make a good contribution to Proc Roy Soc B.

Appendix A

Response to referees

We thank the reviewers for their thorough and thoughtful reviews. All criticisms are addressed in point-by-point fashion below. To compensate for the increase in the word count, which is under the strict regulation of the journal style, we decided to move Table 1 and the rest of Materials & Methods to Supplementary Information. All edits in response to the criticisms and a few other corrections were indicated by Track Changes in the manuscript Word file. We believe this has resulted in an improved manuscript that is appropriate for a *Proceedings of the Royal Society B* audience.

Referee: 1

1-0. Comments to the Author(s)

Fukushima et al. demonstrate that the heterophyllous Western Australian pitcher plant (*Cephalotus follicularis*) exhibits a maladaptive reaction norm when exposed to combinations of growth temperature and daylength that occur rarely in the wild. Their meticulous analysis shows that a clear switch from flat non-trap leaves to cylindrical trap leaves dependent on temperature, but a range of abnormal leaf phenotypes when the conditioning temperature and daylength are in combinations seen only rarely – in fact, once in a lifetime, based on the authors' analysis of climatic data from part of the range of *Cephalotus*. This is a brilliant paper, and should be of interest to ecologists, evolutionary biologists, and developmental biologists.

We thank the reviewer for their helpful comments and appreciation of the major finding of this work.

I had three minor suggestions that the author should consider in revising their manuscript:

1-1. The statement on lines 107-108 ("There was no indication that their morphology was more suitable for photosynthesis or carnivory than flat or pitcher leaves, respectively.") and its echo in the Discussion are rather backhanded. I appreciate the authors' careful attempt to not claim what they have not demonstrated. But I do wonder whether a more positive statement – something along the lines that intermediate leaves are unlikely to perform as well at either photosynthesis or prey capture as the naturally occurring flat or trap leaves – might convey their conclusion more clearly.

Response: We agree that the phrasing the reviewer suggested is better and more precise for describing our results. We followed this suggestion.

Change in Results: On the basis of their morphology, the malformed leaves are unlikely to perform as well at either photosynthesis or prey capture as the flat or pitcher leaves.

Change in Discussion: *Cephalotus* produced diverse shapes of malformed leaves (Fig. 1) that are unlikely to perform as well at either photosynthesis or prey capture as the flat or pitcher leaves, suggesting a maladaptive response.

1-2. The authors might want to cite Givnish et al. 1984 (already in the bibliography) or Givnish et al. 2018 in Ellison and Adamec) explaining that some carnivorous plants produce non-trap leaves during unfavorable cool or dry seasons when prey density or the photosynthetic benefits of carnivory are likely to be low. In the case of *Cephalotus* inhabiting the (permanently wet) reed swamps near Albany, summer comes with greater prey density and light availability, which tends to explain the production of trap leaves under those conditions, and the production of non-trap leaves under the cooler, cloudier, shorter days of winter.

Response: We thank the reviewer for letting us know about the latest review article as well as the original, epic paper describing the cost/benefit model of plant carnivory. In response to this comment, we discussed this topic in Introduction and in a relevant Results section.

Change in Introduction: In carnivorous plants, many species are known to produce non-trap leaves during unfavorably cool or dry seasons when the photosynthetic benefits of carnivory are likely to be low (Givnish et al., 1984, 2018).

Change in Results: The cost–benefit model of the evolution of carnivorous lifestyles predicts that abundant light enables the plant to compensate for the cost of trap leaf production by enhancing photosynthesis alongside the acquisition of prey-derived nutrients (Givnish et al., 1984, 2018; Pavlovič and Saganová, 2015). Because such a compensation cannot be accomplished in light-limited conditions, seasonal changes in light availability may explain the evolution of seasonal heterophylly in carnivorous species, including *Cephalotus* (Givnish et al., 1984, 2018).

1-3. The authors might also want to read Givnish 2002 (Evolutionary Ecology) on how the breakdown of a regional correlation of environmental cues that direct plasticity in a given set of traits might impose a deep, unavoidable limit on the extent to which selection can drive an expansion in species range. In the winter rainfall/Mediterranean climate region of SW Western Australia, variation in daylength vs. rainfall and temperature are more or less perfectly inversely and directly related, respectively. The developmental program elucidated by the authors would tend to behave in an adaptive sense within this region. But as one moves outside that region (or simply outside the reed swamp topographic habitat that provides abundant water year-round), the correlations among the environmental signals – which also help determine the net returns from carnivory – break down. This could help limit the range of *Cephalotus* to the SW corner of Western Australia, where the regional correlation of environmental signals remains intact and produces adaptive shifts in leaf morphology in the reed swamp habitat.

Response: We thank the reviewer for pointing out this important aspect. We agree that it is directly relevant to our findings. This point is now discussed in a new Supplementary Text 2 shown below. We first tried to include this paragraph in the Discussion, but this was not possible because of an exceeded word count judged by the online submission system.

Change:

The regional correlation of environmental factors in relation to water availability. Among the environmental factors other than temperature and photoperiod, water availability is of particular interest in relation to the cost-benefit model of plant carnivory, which predicts water

and light availabilities as key factors in the phenotypic plasticity of carnivorous plants (Givnish et al., 1984, 2018). A typical habitat of *Cephalotus* is a peat-soil swamp in permanently damp seepage areas (Cross et al., 2019), and therefore water is available throughout the year without limitation. In such an environment, photoperiod tends to correlate positively with the photosynthetic benefit resulting from carnivory (Givnish et al., 1984, 2018). However, the winter-rainfall climate of the south-western corner of Western Australia inverses such relationships outside the water-rich regions (i.e., negative correlation between photoperiod and water availability). This fact highlights an attractive hypothesis that the breakdown of a regional environmental factor correlation (Givnish, 2002) may compromise an appropriate response by *Cephalotus*, if water availability or its covariate is not well integrated as a cue modulating the phenotypic plasticity.

Referee: 2

2-0. Comments to the Author(s)

Review of A discordance of seasonally covarying cues unmasks a hidden reaction norm in the heterophyllous pitcher plant *Cephalotus follicularis*

I started reading this manuscript with great interest. The study system is fascinating, and the idea that novel environments may induce the production of “strange” or malformed phenotypes even when these novel environments are combinations of factors that the plants experience in the field is really intriguing.

We thank the reviewer for their helpful comments and appreciation of the major finding of this work.

2-1. However, I had a serious problem with the way the manuscript is laid out (I find the contents of the different sections very disproportionate), and importantly, with the methodological description of what actually was done. The methods section in the main text is reduced to one short paragraph. I understand that not all details may fit into the main text, but this is extremely short and lacks all details. I've checked other manuscripts in the journal, and it does not seem to be a journal limitation. Even so, this would be OK if the supplementary methods were thorough. However, I also found this material confusing and incomplete.

Response: We agree that Materials and Methods should be detailed enough for the reproducibility and proper data interpretation. In the original manuscript, we presented a large part of Materials and Methods in Supplementary Information because of the need for the complete description of our results under the strict page limitation in *Proceedings of the Royal Society B*. In response to this comment, we provided all missing information including those previously presented in a less-accessible format. Some confusion may have arisen because the method has split into the main and supplemental texts, making it difficult to quickly locate the required information. In order to meet both the page limitation and searchability, we decided to move the entire parts of Materials and Methods to Supplementary Information except for the data availability statement. Point-by-point responses are provided below.

2-2. First, there is no explanation of the genetic material that was used. There is only a short sentence stating that the plants were obtained in a nursery. However, the introduction states that almost 4000 genetically-identical individuals were used. This is very confusing. Are they all clones? What is the origin of this material? Do they come from the same populations? How can the reader know if the experimental treatments are “normal” for these plants if we don’t know their origin? Also, looking at the map of the distribution of the species, it is likely that different populations experience different environmental conditions, and hence, different combinations of abiotic factors. Then, justifying the origin of the material and the climatic conditions of the study material is critical.

Response: We thank the reviewer for this important comment. In response to this comment, we provided the details of plant materials as follows.

Change:

Plant materials. Because it is impractical to obtain a large number of individuals (>3,000 plants) from wild populations of *Cephalotus*, which is currently listed as a “vulnerable” species on the IUCN Red List of Threatened Species (Bourke et al., 2020), we used a vegetatively propagated axenic culture strain (i.e., in vitro plants), which we obtained previously from CZ Plants Nursery (Trebovice, Czech Republic) for the genome sequencing (Fukushima et al., 2017). The geographical origin of this strain is unknown, and therefore, we discussed the results of our laboratory experiments by taking into account the meteorological similarities and differences in four representative localities that cover the distribution limits of *Cephalotus* (see “Meteorological data” and Fig. S2). The plants were maintained in polycarbonate containers (60 × 60 × 100 mm) containing half-strength Murashige and Skoog medium (Murashige and Skoog, 1962) supplemented with 3% sucrose, 1× Gamborg’s vitamins, 0.1% 2-(N-morpholino)ethanesulfonic acid, 0.05% Plant Preservative Mixture (Plant Cell Technology), and 0.3% Phytigel (Fig. S9). The plants were vegetatively propagated by shoot cuttings. Newly propagated plants were supplied for experiments (i.e., plants were not reused for multiple experiments).

2-3. My second concern is the lack of clarity in the explanation of the experimental treatments and the experiments themselves. Is this just one experiment where light and temperature were manipulated? Is it different experiments? I’ve looked through Table S1, where the experiments are supposedly detailed, and I couldn’t make sense of it.

Response: We performed 18 rounds of 12-weeks growth experiments spanning 7 years. To better present the experimental designs, we generated a pivot table summarizing growth conditions, the period of experiments, and the number of plants used in each experiment. This is presented as new Table S3 in the revised manuscript. The code to generate new Table S3 from new Table S2 (previous Table S1) is attached as a Supplementary Data file on figshare (“20200925_experimental_design.ipynb”). Furthermore, we provided in the same Excel file the legend of Supplementary Tables for detailed descriptions of the variables presented in new Table S2.

Change:

Table S3. The number of plants used in experiments.

Category of experiments			Temperature-photoperiod combinations															
Temperature of preculture			25.0°C	25.0°C	25.0°C	25.0°C	25.0°C	25.0°C	25.0°C	25.0°C	25.0°C	25.0°C	25.0°C	25.0°C	25.0°C	25.0°C	25.0°C	25.0°C
Temperature of main culture			15.0°C	15.0°C	15.0°C	17.5°C	20.0°C	20.0°C	20.0°C	22.5°C	22.5°C	22.5°C	25.0°C	25.0°C	25.0°C	25.0°C	25.0°C	25.0°C
Photoperiod			16L8D	24L0D	8L16D	24L0D	10L14D	12L12D	24L0D	10L14D	12L12D	24L0D	10L14D	12L12D	14L10D	16L8D	24L0D	8L16D
Leaf categorization			Fully categori	Fully categori	Fully categori	Fully categori	Fully categori	Fully categori	Fully categori	Fully categori	Fully categori	Fully categori	Fully categori	Fully categori	Fully categori	Fully categori	Fully categori	Fully categori
Start (YYYYMMDD)	End (YYYYMMDD)	Place	Figure	Number of plants														
20120712	20121016	NIBB	Fig. S8	0	0	0	0	0	0	0	0	0	0	0	0	0	0	0
20120719	20121015	NIBB	Fig. S8	27	25	31	0	0	0	0	0	0	0	0	0	0	0	0
20120720	20121015	NIBB	Fig. S8	0	0	0	0	0	0	0	0	0	0	0	0	0	0	0
20120723	20121210	NIBB	Fig. 2B,C	0	0	0	0	0	0	0	0	0	0	0	0	0	0	0
20130405	20130628	NIBB	Fig. S4	0	0	0	0	0	0	0	0	0	0	0	0	0	0	0
20131016	20140108	NIBB	Fig. 2F; Fig. 3; Fig. 4A	0	0	0	0	0	0	0	0	0	0	0	0	0	45	45
20140123	20140417	NIBB	Fig. 2F; Fig. 3; Fig. 4A; Fig. S8	0	45	0	45	0	0	45	0	45	0	0	0	0	0	90
20140328	20140828	NIBB	Fig. 2F; Fig. 3; Fig. 4A	0	45	0	45	0	0	45	0	0	45	0	0	0	0	90
20141017	20150113	NIBB	Fig. 2F; Fig. 3; Fig. 4A; Fig. 4B	0	45	0	0	0	0	0	0	0	0	0	45	0	45	45
20180426	20180719	NIBB	Fig. 4B	0	0	0	0	0	0	0	0	0	0	0	17	0	0	0
20181016	20190108	NIBB	Fig. 4B	0	0	0	0	45	0	0	45	0	0	0	0	0	0	0
20190108	20190402	NIBB	Fig. 4B	0	0	0	0	0	0	0	0	0	0	36	0	36	0	0
20190111	20190405	NIBB	Fig. 4B	0	0	0	0	0	0	0	0	0	0	9	0	9	0	0
20190221	20190516	NIBB	Fig. 4A	45	0	0	0	0	0	0	0	0	0	0	0	0	0	0
20190228	20190523	NIBB	Fig. 4A	0	0	48	0	0	0	0	0	0	0	0	0	0	0	0
20190412	20190705	NIBB	Fig. 4B	0	0	0	0	0	0	0	0	0	45	0	45	0	0	0
20190708	20190930	NIBB	Fig. 4B	0	0	0	0	0	45	0	0	45	0	0	0	0	0	0
20190801	20191107	Univ Würzburg	Fig. 4B	0	0	0	0	95	0	0	91	0	0	0	0	0	0	0

(a part of Table S3 is shown here, see the Excel file for the full details)

Legends of Supplementary Tables.

Table S1. Classification of leaf morphology in *Cephalotus*.

Table S2. The heterophylly data analyzed in this study.

experiment Unique experimental code.

start Start date of main culture in YYYYMMDD.

end End date of main culture in YYYYMMDD.

period Days of main culture.

operator Name of the person who conducted the experiment.

medium Culture medium. HMS = half-strength MS-based medium.

temperature Temperature of main culture (°C).

photoperiod Photoperiod of main culture. e.g., 16L8D means 16-h day and 8-h night.

complex_conditions Different daytime and nighttime temperature in combinations with daylength.

light_quality Light quality.

light_intensity_lower Lower limit of light intensity ($\mu\text{mol}\cdot\text{m}^{-2}\cdot\text{s}^{-1}$).

light_intensity_upper Upper limit of light intensity ($\mu\text{mol}\cdot\text{m}^{-2}\cdot\text{s}^{-1}$).

lamp Type of light source.

seal Materials used for the sealing of polycarbonate containers.

treatment Other experimental categories. Week = weeks of growth

treatment_value Associated values of "treatment". In this study, this column is used only for indicating weeks of growth.

order How leaf symbols are ordered in "heterophylly". For example, if "YoungToOld", Ps are younger than Fs in "PPPPFF".

plant_id Unique IDs of individuals within the experimental replication.

heterophylly Heterophyllous shifts of leaf categories in the main shoot.

intermediate How malformed leaves were recorded. See Materials and Methods for details.

count_all_leaves Whether all leaves in the main shoots are recorded.

comment Miscellaneous information.

Table S3. The number of plants used in each experiment.

Place NIBB, National Institute for Basic Biology, Japan; Univ Würzburg, University of Würzburg, Germany

Figure Figures in which data were used.

2-4. Furthermore, I was not find any information on how the experiments were actually performed. Were they done in a greenhouse? On growth chambers? How long did they last? Were plants potted individually? I'm just very surprised by the lack of detail of the methods section, and puzzled about how this could be at all reproducible.

Unfortunately, the lack of clarity on the experiments makes it impossible to put the results in perspective, and prevents from drawing meaningful conditions. I'm sorry I cannot be more positive.

Response: In response to this comment, we described the use of growth chambers. The other requested information is available in the same Method section which was originally presented as a part of the main text and now moved to the Supplementary Information in the revised version (below).

Change:

Analysis of heterophylly. To examine plant phenotypes under controlled and constant environments, we used aseptic cultures for the experiments. Plants were precultured for months at 25°C in continuous light. A preculture at 15°C did not substantially change the pattern of heterophyllous phenotypes in continuous light (Fig. S8). To ensure uniform plant conditions at the onset of the experiments, we selected asexually produced shoots with one or two expanded leaves and transferred them to a new medium. Paraffin-embedded tissue sections of a typical shoot tip were prepared as described previously (Fukushima et al., 2015), and are illustrated in Fig. S9, where the youngest two leaf primordia do not show signs of flat- or pitcher-leaf morphogenesis. Either nine or 16 plants were allocated to each container. The plants were then grown for approximately 12 weeks, and subsequently, we recorded the morphology of the produced leaves and their phyllotactic positions in the main shoots. Leaves on axillary shoots were omitted. **All experiments were conducted in growth chambers (MIR-150, MIR-153, MIR-154, and MLR-350 by PHC, CLE-405 by TOMY SEIKO, LP-1PHS by Nippon Medical & Chemical Instruments Co., Ltd., PK 520-LED by poly klima GmbH, and their equivalents) with controlled temperature and daylength (Fig. S9).**

2-5. Line 28: what do you mean by “normal”?

Response: This “normal” is used in a relative sense to contrast the “extreme” in the next sentence connected by a semicolon. We have discussed whether this word can be paraphrased, but concluded that it was difficult to do so without destroying the flow of the explanation under the strict word limitation in the Abstract. Therefore, we decided to present it as is, while providing in the main text a quantitative interpretation of normal and extreme conditions with *Cephalotus* as an example (see Fig. 4C–D and relevant discussion in the main text).

2-6. L30: does a hidden norm of reaction always involve the mis-regulation of phenotypes?

Response: To our knowledge, whether they are always linked or not is an unsolved question, so we would like to maintain the current form of presentation not to overstate our understanding. Although a citation is not possible in Abstract (L30), we navigate in Introduction the readers to Schlichting (2008) for a detailed review of relevant topics.

2-7. I would like to see some more specific methodological details included in the abstract. It should be a stand-alone document, and in current state, it is missing key information.

Response: We thank the reviewer for this important suggestion. We agree that more methodological clarification would improve the Abstract. In response to this comment, we have changed the Abstract as follows.

Change: To elucidate the combinatorial effects of environmental factors, we studied the leaf shape plasticity of the carnivorous pitcher plant *Cephalotus follicularis* ~~in controlled environments~~. ~~Clonally propagated plants were subjected to 12-week-long growth experiments in different conditions controlled by growth chambers~~. Here, we show that the dimorphic response of forming a photosynthetic flat leaf or an insect-trapping pitcher leaf is regulated by two covarying environmental cues: temperature and photoperiod.

2-8. L50-51: is all plasticity driven by genetic assimilation and accommodation?

Response: To our knowledge, there is no evidence supporting the link between all plasticities and genetic assimilation/accommodation. As we have not implied the universal relationship, we would like to maintain the current presentation not to go beyond what is already known.

2-9. L97-98: I find this phrasing odd. Plasticity is a property (the slope) of the reaction norm. What do you mean by “a hidden reaction norm in phenotypic plasticity”?

Response: Thank you for pointing this out. We agree with this comment. In response to this comment, we edited as follows.

Change: We show that an unusual combination of temperature and photoperiod unmasks a hidden reaction norm ~~in phenotypic plasticity~~, even though neither environmental cue is extreme.

2-10. L376: do you mean there was no dark period?

Response: Yes, continuous light means 24L0D. To further clarify this, we edited the sentence as follows.

Change: Plants were precultured for months at 25°C in continuous light (24L0D).

2-11. Supplementary methods

A description of the species and the type of habitats where it occurs would be desirable.

Response: In response to this comment, we revised Supplementary Methods. For detail, please see the response to #2-2.

2-12. L666: what is axenic culture? Please describe.

Response: In response to this comment, we revised Supplementary Methods. For detail, please see the response to #2-2. In addition, we included a new Supplementary Figure to visually show a setup for growth experiments (new Fig. S9).

Change:

The plants were maintained in polycarbonate containers (60 × 60 × 100 mm) containing half-strength Murashige and Skoog medium (Murashige and Skoog, 1962) supplemented with 3% sucrose, 1× Gamborg's vitamins, 0.1% 2-(N-morpholino)ethanesulfonic acid, 0.05% Plant Preservative Mixture (Plant Cell Technology), and 0.3% Phytigel (Fig. S9).

Fig. S9. Growth experiment in a climate chamber. (A) A front view of a growth chamber. **(B)** A lateral view of a plant container. **(C)** A top view of a plant container. Nine *Cephalotus* plantlets were grown in a medium.

2-13. L667: where do these plants come from? Are they from the same provenance?

Response: In response to this comment, we revised Supplementary Methods. For detail, please see the response to #2-2.

2-14. L681-683: not clear what this sentence adds. Please explain.

Response: In response to this comment, we added a description as follows.

Change: When combined with an appropriate day length, the use of daily mean temperature reproduces a key component of the seasonal transcriptomes in *Arabidopsis halleri* subsp. *gemmifera* (Nagano et al., 2019), suggesting that constant temperatures are a reasonable approximation of daily fluctuating temperatures (see Supplementary Text 1 for further discussion).

2-15. L685: what do you mean by benign?

Response: This “benign” can be paraphrased with “naturally occurring”. To prevent a misunderstanding, we paraphrased it as follows.

Change: We also examined a range of naturally occurring photoperiods (10L14D, 12L12D, and 14L10D).

2-16. L668-687: the experiments need to be described clearer and provide much more detail. I've looked through Table S1 and it is not clear if this was just one experiment or several experiments. Was there a first experiment comparing short vs long day (L684) at a single temperature? And then a second experiment with diverse day lengths?

Response: Yes, we first examined short vs long day at 15°C (Jul 2012 to Oct 2012) and 25°C (Oct 2013 to Jan 2014) and then started analyzing the other day lengths. Later we performed more experiments to check the reproducibility. Because it is too complicated to explain by text, we summarized it in new Table S3. For details, please see the response to #2-3. We also added citations to new Table S3 where new Table S2 (previous Table S1) was originally referred to as follows.

Change: In most experiments, 32 or 45 plants were examined for heterophylly (Table S3). We performed the experiments one to five times (Table S3) and presented the pooled data because consistent results were obtained among the individual experiments.

2-18. How were these replicates (1-5) done?

Response: This information has been provided in Table S3 in the revised manuscript. In response to this and other comments, we added Table S3. For details, please see the response to #2-3.

2-19. With the same individuals?

Response: In response to this comment, we added an explanation as follows in the Methods section “Plant materials”.

Change: Newly propagated plants were supplied for experiments (i.e., plants were not reused for multiple experiments).

2-20. Where were the experiments done?

Response: The location info is now provided in Table S3. In response to this and other comments, we added Table S3. For details, please see the response to #2-3.

2-21. In a greenhouse, growth chamber?

Response: We used growth chambers throughout this work. In response to this and other comments, we revised Supplementary Methods. For details, please see the response to #2-4.

2-22. Were plants potted individually?

Response: Either nine or 16 plants were allocated to each container. This has been addressed in #2-4.

2-23. And most importantly, what is the genetic structure of the sample? Do all individuals come from the same genetic origin? Are they clones? The end of the introduction mentions genetically-identical individuals, but I can't find a proper description of the structure of the experimental plants.

Response: We used clonally propagated plants. This has been addressed in #2-2.

2-27. L694: but what experiments are these? Why was this categorization used for one or other experiments?

Response: These differences stem from a historical reason. The binarized scheme had been used in the early phase of this study before we unexpectedly found the fascinating aspect of malformed leaf production. At that time, nobody was aware of the fact that the malformed leaves were inducible (a major finding of this paper), and most leaves we observed were either flat or pitcher. Therefore, we used the binarized scheme to deal with the infrequent production of malformed leaves. In compiling this paper, we removed most of the early-phase data in which the binarized scheme was used, to highlight the malformed leaf production, but decided to show a part of them when the binarized scheme does not compromise the data interpretation (i.e., growth curve in Fig. 2B,C and effect of preculture temperatures in Fig. S8). Nevertheless, all major findings have been supported by the “fully categorized” scheme. In response to this comment, we summarized the use of leaf categorizations of each experiment in Table S3 and cited it accordingly.

Change: The binarized scheme was used only in the experiments presented in Fig. 2B,C and Fig. S8, while the fully categorized scheme was used for all other experiments (Table S3).

2-28. L699: what do you mean by this? are you referring to the center of the distribution? Is it where the species is most abundant?

Response: In response to this comment, we added an explanation as follows.

Change: The Albany region has been recognized as a representative habitat because of abundant specimen records (Fig. S2) since its discovery within this region in the early 1800s (Cross et al., 2019).

2-29. Figure 1: I'd suggest numbering the panels instead of using letters. Because the leaves are categorized also by letters, it's just confusing.

Response: To strictly follow the journal style, we maintained the letters that indicate sub-panels. However, we agree that we should prevent unnecessary confusion. To better discriminate the subpanel letters and leaf categories, we edited the labels: e.g., “P-type leaves” rather than just “P”. The same change has been applied to Fig. S1.

Change in Fig. 1:

Change in Fig. S1:

Appendix B

Response to referees

Referee: 2

I appreciate the author's efforts to improve the clarity of the methods. I find this revision much more improved, and it is now possible to make sense of the experiments that were done and the interpretation of the results.

Response: We thank the reviewer for another round of their thorough and thoughtful reviews. All criticisms are addressed in point-by-point fashion below. All edits in response to the criticisms and a few other corrections were indicated by Track Changes in the manuscript Word file.

In all honesty I have to say that I find some of the methodological choices of the authors somewhat problematic (e.g. the unknown origin of the material), and some of the methodological aspects of the experiment were still unclear.

Response: In response to this comment, we detailed in Supplementary Methods the species identification and the possibility to trace the geographical origin of the cultivated strain.

Change: Although the geographical origin of this strain is unknown, a species misidentification is unlikely because the genus *Cephalotus* and the family Cephalotaceae are both monotypic (i.e., *C. follicularis* is the only extant species) [12]. As whole-genome sequences are already available [15], molecular genetic studies on the geographical differentiation of this species will reveal the origin of the cultivated strain we used. Throughout the manuscript, we discussed the results of our laboratory experiments by taking into account the meteorological similarities and differences in four representative localities that cover the distribution limits of *Cephalotus* (see "Meteorological data" and Fig. S2).

For instance, I could not find the light levels on the growth chamber, which usually are on the low side. I wonder how this might have affected the combination of light and temperature and whether these combinations were "normal" or not. I nevertheless find the results interesting, and leave to the editor whether the revision meets the standard for publication.

Response: Unfortunately, we do not have data on light intensity in the natural habitats, but we previously checked its effect in a growth experiment. The results are now described in Supplementary Methods and visualized in the new Fig. S9.

Change in Supplementary Methods: We also examined the effect of light intensity ranging 8 to 54 $\mu\text{mol m}^{-2} \text{s}^{-1}$ under continuous light. Because no substantial effect was observed (Fig. S9), the rest of the experiments were conducted under a light intensity of 20 to 40 $\mu\text{mol m}^{-2} \text{s}^{-1}$.

New Fig. S9:

Fig. S9. The effect of light intensity. Heterophylly in different combinations of temperature and light intensity in continuous light (24L0D). The heterophyllous leaf production was recorded after 12 weeks of the main culture. Note that malformed leaves were categorized into flat-like or pitcher-like in this analysis (i.e., “binarized” scheme, see Supplementary Methods and Table S1). The flat-like category represents categories F, C, H, M, and I, whereas the pitcher-like category includes categories P, L, and D. Although fully categorized data were not obtained, the proportion of malformed leaves was stably low. The numbers of observed leaves (N) are indicated above the stacked bar plots.